# CD24 Is a Prognostic Marker for Multiple Myeloma Progression and Survival

**DOI:** 10.3390/jcm11102913

**Published:** 2022-05-20

**Authors:** Noa Gross Even-Zohar, Marjorie Pick, Liron Hofstetter, Adir Shaulov, Boaz Nachmias, Eyal Lebel, Moshe E. Gatt

**Affiliations:** 1Department of Hematology, Hadassah Medical Center, Faculty of Medicine, Hebrew University of Jerusalem, Jerusalem 9112102, Israel; marjorie@cc.huji.ac.il (M.P.); adir@hadassah.org.il (A.S.); boazn@hadassah.org.il (B.N.); eyal.lebel@gmail.com (E.L.); rmoshg@hadassah.org.il (M.E.G.); 2Institute of Hematology, Davidoff Cancer Center, Rabin Medical Center, Petah Tikva 4941492, Israel; lironhof@gmail.com; 3Sackler Faculty of Medicine, Tel Aviv University, Tel Aviv 6997801, Israel

**Keywords:** multiple myeloma, prognosis, CD24, multiparameter flow cytometry

## Abstract

Surface antigens are commonly used in flow cytometry assays for the diagnosis of multiple myeloma (MM). Some of these are directly involved in MM pathogenesis or interactions with the microenvironment, but most are used for either diagnostic or prognostic purposes. In a previous study, we showed that in-vitro, CD24-positive plasma cells exhibit a less tumorigenic phenotype. Here, we assessed the prognostic importance of CD24 expression in patients newly diagnosed with MM as it correlates to their clinical course. Immunophenotyping by flow cytometry of 124 patients uniformly treated by a bortezomib-based protocol was performed. The expression of CD24, CD117, CD19, CD45, and CD56 in bone marrow PCs was tested for correlations to clinical parameters. None of the CD markers correlated with the response rates to first-line therapy. However, patients with elevated CD24+ expression on their PCs at diagnosis had a significantly longer PFS (*p* = 0.002) and OS (*p* = 0.044). In contrast, the expression of CD117, CD56, or CD45 was found to have no prognostic value; CD19 expression was inversely correlated with PFS alone (*p* < 0.001) and not with OS. Thus, elevated CD24 expression on PCs appears to be strongly correlated with survival and can be used as a single-surface antigenic prognostic factor in MM.

## 1. Introduction

Multiple myeloma (MM) is characterized by a neoplastic proliferation of clonal plasma cells (PCs). These clonal PCs produce immunoglobulins and cause typical end organ damage such as anemia, kidney failure, lytic bone lesions, and hypercalcemia [1,2,3]. Over the last two decades, important progress has been made in understanding the pathophysiology of MM. This has enabled significant advances in our ability to treat patients with MM using novel agents that can prolong life expectancy and quality of life. Nevertheless, MM remains a largely incurable disease [4,5]. Most patients with MM have an asymptomatic pre-malignant state of monoclonal gammopathy of unknown significance (MGUS) and/or smoldering MM (SMM). However, many go undetected until active disease is apparent [1]. Various factors influence malignant transformation, including primary genetic events, clonal evolution dependent on secondary genetic events, bone marrow microenvironment (BMM) changes, and the failure of the immune system to eradicate the malignant clone [6].

The BMM milieu is composed of different components, including, most notably, cellular elements such as osteoblasts and dendritic cells; the extracellular matrix, including collagen, proteoglycans, and hemonectin; and soluble elements such as cytokines, growth factors, and adhesion molecules. The BMM sustains a close interaction with the hematopoietic stem cells (HSCs) and provides the necessary support for hematopoiesis. In addition, the BMM is an essential part of the malignant transformation to MM through a variety of interactions with the malignant PCs [7]. One of the factors that is known to interact with the BMM is CD24.

CD24 is a glycoprotein typically found on the membrane surface. It is thought to stimulate cell adhesion to the BM extracellular matrix [8]. It acts as a differentiation marker on B-cell progenitors, and its expression decreases once lymphocytes enter the germinal center and undergo terminal differentiation [9]. High CD24 expression in breast, prostate, pancreas, ovary, colorectal, and bladder tumors was shown to be correlated with metastatic disease and poor prognosis [10]. The important role of CD24 protein expression in MM in vitro was confirmed by our group. CD24+ MM PCs were characterized by a less tumorigenic phenotype than CD24− MM PCs [11]. Previous publications have shown that CD24 mRNA expression is correlated with good prognosis [12]. To the best of our knowledge, no studies have examined the role of CD24 in primary amyloidosis (AL). 

Multiparameter flow cytometry (MFC) immunophenotyping remains the mainstay in the diagnosis and monitoring of PC dyscrasias. Aside from its crucial role in diagnosis, MFC plays a significant role in monitoring responses to treatment and assessing minimal residual disease (MRD) status [13]. At diagnosis, a few specific surface antigens measured by MFC can identify the pathogenic clone and further stratify it. Some markers, such as CD19, CD117, CD45, and CD56, were found to have prognostic significance [14,15,16,17,18,19]. 

Taken together, we hypothesized that CD24 expression, as measured by flow cytometry on PCs of patients newly diagnosed with MM, would have significant prognostic value.

## 2. Materials and Methods

### 2.1. Patient Samples

The clinical data of 124 patients newly diagnosed with multiple myeloma (MM) or primary amyloidosis (AL) at Hadassah Medical Center, Jerusalem, Israel and treated between September 2011 and June 2017 were retrospectively collected. To be eligible, patients had to be uniformly receiving first-line bortezomib-based treatment for at least one cycle. For purposes of uniformity, patients treated with other regimens were excluded. 

To determine the expression patterns relative to disease stage measurements, the BM of patients was examined and analyzed during their routine follow-up visits. This cohort was separate from the prognostication cohort. The collection of patient data for both cohorts was approved by the local Institutional Review Board.

### 2.2. Data Collection

The clinical data included baseline patient and disease characteristics at diagnosis, treatment protocol, response to treatment, and long-term outcomes, which were defined as progression-free survival (PFS) and overall survival (OS). The diagnoses of MM and AL were based on the International Myeloma Working Group criteria for the diagnosis of MM [2]. High-risk cytogenetics were considered based on the presence of del (17p) and/or translocation t (4;14) and/or translocation t (14;16) following the R-ISS criteria [20]. All patients with AL were tissue-proven by positive Congo-red stain and immunohistochemistry typing. Further electron microscopy immunogold typing was utilized in 7 out of 19 patients. Response assessment was performed according to the International Myeloma Working Group [21] and AL consensus statement criteria [22].

### 2.3. Immunophenotyping of the Samples

Sample analysis was performed using 4-color staining with a FACScalibur (BD Bioscience, Palo Alto, CA, USA). For each sample, antibodies were added to 75 µL of whole BM (FITC/PE/PerCP/APC) in the following combinations: 1. (CD38/CD117/CD45/CD19), 2. (CD38/CD24/CD45/CD56), and 3. (CD38/CD81/CD45)/CD56). All antibodies were purchased from Beckman Coulters. After adding the antibodies, the test tubes were placed for 20 min at 4 °C in the FACS. Erythrocytes were lysed by incubating the samples with 2 mL of lysing solution (BD Bioscience) for 5 min at room temperature. Next, the cells were pelleted by centrifugation for 4 min at 1500 RPMI. The cells were re-suspended and 300 µL of phosphate-buffered saline was added to each test tube. Samples were acquired using the FACScalibur flow cytometer and CellQuest pro program (BD Bioscience). PCs were gated according to their side scatter and high expression of CD38. A minimum of 3000 CD38+ BM cells were acquired. Myeloma PCs were differentiated from normal PCs based on the previously described work of Paiva et al. [23] and in line with the recommendations of the European Myeloma Network for cell expression of CD117, CD19, CD45, and CD56 [24]. 

### 2.4. Statistical Analysis

Kaplan–Meier survival analysis, with the log rank test for the comparison of survival curves, was applied to assess the effect of the categorical variables on overall survival and progression-free survival. The effects of the quantitative variables on overall survival and progression-free survival were tested using the Cox regression model. The Cox regression model was also used as the multivariate model for simultaneously testing the effects of several variables on overall survival and progression-free survival. This model was applied using the stepwise, forward, likelihood ratio method. The comparison of quantitative variables for three or more independent groups was carried out by applying the Kruskal–Wallis (K–W) non-parametric ANOVA test. Pairwise comparisons (post hoc tests) following the K–W test were performed using the Mann–Whitney non-parametric test with the Bonferroni correction of the significance level. Non-parametric tests were implemented because of the small sample sizes and non-normal distributions of the variables in some of the groups. All tests were two-tailed, and a *p*-value of 0.05 or less was considered statistically significant.

## 3. Results

The expression patterns of CD19, CD56, CD117, CD45, and CD24 on the PCs in the BM of patients’ samples at the different clonal evolutionary stages of MM development are presented in Figure 1A–E. As shown, the patterns were different for each marker. Whereas the CD56 and CD117 levels increased between the MGUS and MM diagnosis and then remained stable during further active disease evolution, the opposite was observed for the CD45 pattern. CD24 expression levels were generally lower than the other CDs on PCs during the evolutionary development of the cells and were similar to the pattern seen for the early B-cell marker CD19 (Figure 1F). Unlike CD19, CD24 levels were elevated in the progressive disease stage. Thus, these results suggest that each marker has its own independent dynamics.

The main cohort of 124 patients was assessed for prognostic purposes. Their baseline characteristics, separated according to CD24+/− and AL/MM, are summarized in Table 1. 

There was a higher prevalence of affected males (60.5%) than females (39.5%), and the median age was 62.5 years (range 30–89). Of all patients, 105 (84.7%) were diagnosed with MM and 19 (15.3%) with AL. Almost all (96%) of the patients were receiving cyclophosphamide, bortezomib, and dexamethasone (VCD) as the induction treatment. The remaining patients (4%) were treated with a bortezomib, thalidomide, and dexamethasone (VTD) regimen. Forty-six percent were treated with autologous stem cell transplantation (ASCT) as consolidation. There were no correlations between any of the CD expression levels and the patients’ baseline characteristics (not shown). There were no differences between patients with MM and AL for any of the CD expression levels, or for OS or PFS (Table 2 and Table 3, and Appendix A Appendix A).

Note that there were no statistical differences between the patients with MM and AL for any of the analyses; all were treated by a similar bortezomib-based protocol. Moreover, all outcomes were similar in both groups of patients. Therefore, to enhance the statistical power of the cohort, results are presented separately in the results presented in the next sections. However, when drawing conclusions, they are treated as one group.

Most patients responded to the first-line bortezomib-based treatment. The overall response rate (ORR) was 86.3%, with a very good partial remission (VGPR) rate of 37.7% and a complete remission (CR) rate of 18.5%. No correlation was found between the depth of the response and any of the CD expression levels analyzed, including CD24 (not shown). 

At the median follow-up of 62.2 months (range: 0.4–142.0 months), the median PFS of all patients was 27.5 months (range 0.4–123.1). PFS was significantly correlated with CD24 levels above 5% on PCs at diagnosis in the MM group of patients (*p* = 0.021). In the patients with AL, the trend was clear, with a *p*-value of 0.077. For patients with MM, CD19 was significant as well, with a *p*-value of 0.019. None of the other CDs were significant, and no trends emerged (Appendix A Appendix A). When combining the patients with AL and MM, an expression of CD24 levels above 5% on PCs at diagnosis was significant, with a median PFS of 36.2 months as compared to 22.8 months in those with lower expression levels (*p* = 0.002). CD19 expression was inversely correlated with a PFS of 21.1 months as compared to 35.0 months in those with lower expression levels (*p* = 0.004) (Figure 2). 

This was also observed when assessing the correlation between CD expression levels as continuous variables and PFS. In the MM group, CD24 and CD19 were significantly correlated to PFS (*p* = 0.005 for CD24 and *p* < 0.001 for CD19). In the AL group, neither CD24 nor any of the other CDs were significant. When pooling the patients with MM and AL, both CD24 and CD19 expression levels were found to be significantly correlated with PFS, with a hazard ratio (HR) of 0.975 (*p* = 0.002) and 1.025 (*p* < 0.001), respectively (Table 4). 

None of the other markers, i.e., CD56, CD117, or CD45, showed expression levels that correlated as continuous variables or at the 5% cutoff on PCs with PFS.

The median OS of all patients was 83.5 months (range 0.4–142.1 months). As shown in Appendix A Appendix A, within the MM group, none of the CDs correlated significantly with OS at a cutoff level of 5%. However, CD24 levels >5% showed a trend towards a better OS (*p* = 0.089). AL PC CD24 levels >5% were significantly correlated with a better OS (*p* = 0.017), as were CD56 levels above 5% in this patient group (*p* = 0.039). When pooling the MM and AL cohorts, the only CD reaching near-statistical significance at the 5% cutoff that correlated with OS was CD24 expression on the PCs (*p* = 0.059) with a median OS of 107.5 months as compared to 65.4 months in those with lower expression levels (Figure 3).

Furthermore, when assessing the correlation between CD expression levels as continuous variables to OS, no CD expression reached significance in the MM or the AL groups separately. However, when pooling both cohorts, CD24 was the only CD that correlated significantly with OS (*p* = 0.04); HR 0.979 (Table 5).

The revised international scoring system (R-ISS), which is the standard method for newly diagnosed patient prognostication in MM, was found to correlate with both PFS and OS (HR = 1.917 with a CI of 1.074–3.425, *p* = 0.028, and HR = 2.041 with a CI of 1.003–4.156, *p* = 0.049, respectively). We conducted two multivariate Cox regression analyses. The first included the R-ISS and CD24, and showed that the latter did not retain its significance as an OS predictor (*p* = 0.13), although there was a statistical trend towards significance when the R-ISS and CD24 PFS were assessed (*p* = 0.06; HR = 0.984 with a CI of 0.967–1.001). The second analysis included all variables that were found to be significant in the univariate analysis (Table 6).

The multivariate analysis indicated that two R-ISS components (B2M and cytogenetics), in addition to high calcium levels and older age, were significant. The performance of autologous stem cell transplantation (ASCT) did not correlate with OS (Appendix A Appendix A; *p*-value = 0.262), but when dividing the patients into groups based on who went through ASCT and who did not, a CD24+ status showed a longer PFS (Appendix A Appendix A; *p*-value = 0.012). Other parameters, such as LDH levels, (*p*-value = 0.115), hemoglobin levels (*p*-value = 0.288), the presence of lytic lesions (*p*-value = 0.425), and the presence of PC in bone marrow (0.506), were not found to be significant. When including CD24 levels with all the other significant univariate markers as part of the equation, the *p*-value was 0.383 (HR = 1.011 with a CI of 0.986–1.037).

These parameters were not included in the final multivariate Cox regression analysis since the R-ISS is the acceptable validated standard method for newly diagnosed patient prognostication utilized in everyday practice. 

## 4. Discussion

Although MM is an incurable disease, survival has greatly increased; however, it varies significantly across patients. Methods for classifying prognosis are important at the time of diagnosis and change mostly as a result of progress in available treatments. Although the new standard of risk assessment at diagnosis of active disease is the R-ISS, other modalities such as flow cytometry may contribute to individual patient assessment. This study evaluated the prognostic significance of CD24 expression on PCs in newly diagnosed patients treated uniformly with first-line bortezomib-based therapy.

Multiparameter flow cytometry (MFC) is used in all types of PC disorders for monitoring the malignant load of PCs at diagnosis and for the detection of minimal residual disease (MRD) after therapy [24,25,26]. The phenotypically normal versus aberrant PC ratio, as characterized by MFC at diagnosis, is associated with better OS and PFS [13,27]. There is a relative consensus as to which surface molecules, measured together by flow cytometry, can best identify the pathogenic clone and further stratify it [24,28]. Typically, CD38, CD138, and CD45 are the best backbone markers for the identification of all BM PCs. CD45 is a tyrosine phosphatase which nevertheless has elicited some debate as to its prognostic value. The largest study investigating its role showed it was correlated with the aggressiveness of the tumor and that higher levels correlated with a negative prognostic value [14]. CD117 is a receptor tyrosine kinase that is normally expressed by hematopoietic progenitors in the BM but is absent during B-cell maturation. It is associated with favorable outcomes [17]. The over-expression of CD19, which is part of the B-cell receptor, and CD81, which regulates CD19, were shown to be independent negative prognostic markers [18,29]. CD56 is a membrane glycoprotein expressed on MM cells but not on normal PCs. A lack of CD56 at diagnosis was found to be correlated with poorer prognosis [15,16,19]. However, other studies have not only failed to replicate these findings but also reported contradictory results [30]. Other molecules have been described but are only utilized with next-generation eight- or more-colored MFC [31,32]. The current study focused on the role of CD24 as a prognostic MFC marker at diagnosis. 

We first analyzed CD levels during the different stages of disease evolution and found CD24 to be weakly expressed during the MGUS/SMM stages but up-regulated in the relapsed/resistant stages of disease. Consistent with the claim that CD24 plays a major role in solid tumor metastasis [33], this suggests that MM may lose its stromal dependence as the disease evolves [34,35].

Next, the expression of CD24 on the PCs of the 124 newly diagnosed and uniformly treated cohort of patients with MM and AL was analyzed. None of the CD expression levels assessed, including CD24, correlated with patient baseline characteristics or depth of response to first-line therapy. However, CD24 expression strongly and significantly correlated with both PFS and OS. Regardless of whether we used the multivariate Cox regression analysis and the R-ISS as the standard risk-assessment paradigm for newly diagnosed patients or other significant clinical variables, CD24 did not retain significance as an OS predictor. However, it did exhibit a statistical trend when testing the R-ISS and CD24 for PFS. R-ISS remains a strong predictor and is well-validated [20] but nevertheless has its limitations. R-ISS was developed from data obtained in 11 international, multicenter clinical trials. These clinical trials excluded patients with severe renal dysfunction or with poor performance status, [36] who represent a significant percentage of patients with MM. Furthermore, the tests to calculate the R-ISS (i.e., FISH) are expensive and not universally available [37]. Other known prognostic factors were not included in the R-ISS algorithm [38,39]. In this respect, MFC CD24 remains an important additional modality for assessing prognosis in the patient newly diagnosed with MM. However, it is important to remember that MFC is not a mandatory part of the diagnosis of MM, and that MFC CD24 is not a part of the routinely used or recommended antigen panel used for diagnosis of MM and other plasma cell disorders.

Notably, CD24 levels were the only surface antigen found here to correlate with PFS and OS, although other antigens were previously found to have prognostic significance not seen in the current study. This may be attributed to the relatively small number of patients analyzed. Note as well that previous studies were conducted prior to the era of novel agents, and these prognostic markers may not be as relevant today. These results are in line with our laboratory data showing that CD24+ cells have a less proliferative phenotype [11] and also in line with our previous publication showing that CD24 mRNA expression correlated with good prognosis [12]. Although it examined a smaller series, one study found inverse CD24 correlations with prognosis but provided no data as to which treatments they were exposed to [40]. 

In contrast, the expression of CD117, CD56, and CD45 exhibited no prognostic value. CD19 expression was inversely correlated with PFS, with a HR of 1.025 (*p* < 0.001), but no significant correlation with OS was found. This is in line with CD19 levels previously reported as having a negative predictive value in MM and highlights the differences between CD19 and CD24 [29]. While both are early markers that are largely lost by the time of terminal differentiation of B cells to PCs, their functional role is different, as is their phenotypic role in the tumorigenic process. While CD19 was hypothesized to be a marker of the MM stem cell [41], CD24 takes part, as mentioned above, in the regulation of the critical interaction with the BMM [11,42].

CD24 is a heat-stable, 27-amino-acid-long molecule with a lipid-like structure that binds to the membrane by a glycosylphosphotadylinositol (GPI) anchor. It is thought to function as an adhesion molecule and stimulates cell adhesion [8]. We studied the role of CD24 in the MM PC clones and its relationship with the microenvironment. We found that CD24+ MM cells, as a result of complex interactions with the BMM, migrate less, create fewer colonies, and are considerably more apoptotic as compared with CD24− MM cells; thus, they are less tumorigenic [11]. Even though MM tumor cells, in general, have a high affinity for the BM, the ability of the disease to progress and evolve is tightly connected to interactions between the tumor cells and the BMM. These interactions, including the ones connected to CD24, enable the cells to thrive, migrate out of the BM, and at times develop resistance to anti-tumor therapy [42]. In contrast, Gao et al. described CD24 as a MM-initiating cell marker and found CD24+ cells to have high clonogenic features. However, their experiments did not take the BM stromal cells interactions with MM into consideration [40].

In malignancies, CD24 is found on a variety of solid human tumors. It is believed to play a central role in maintaining cancer cell growth, spreading motility and invasiveness in solid tumors, and its high expression correlates with shorter survival [9,33].

To the best of our knowledge, there are no data on CD24 in AL. The fact that AL, like MM, is a plasma cell disorder treated by similar protocols may lead to a better understanding of the common mechanisms implemented by CD24 that affect the prognosis in AL. We included both groups of patients, although each disease has its own course; however, as shown here, CD expression levels correlated in a similar manner in both, and when pooled, they yielded higher statistical significance. In fact, all the CD levels in the separate cohorts behaved similarly, attesting to the similarities between MM and AL. 

Aside from its prognostic role, CD24 may be a target for new treatments. There are advances in its targeting in ovarian and pancreatic cancers, glioblastoma, and other solid tumors, but minimally in hematologic diseases [43]. Overdevest et al. were the first to study IgG1 mAb mouse anti-human CD24 for the treatment of human bladder cancer cells in mice in 2011; they showed a reduction in tumor growth and metastasis [44]. In MM, Gao et al. used IgG2a mAb mouse anti-human CD24 to treat myeloma cells in immunodeficient mice and showed the inhibition of multiple myeloma cell growth and the prevention of tumor progression [40]. The research in this area is still preliminary. This study has several limitations of note. As is typical of retrospective studies, there were some missing data, and not all sequential patients treated had a full flow cytometry analysis at diagnosis. Patients treated with other induction protocols were excluded. Moreover, the study included patients diagnosed from 2011 to 2017. During this time period, treatment paradigms changed, and newer agents were added to the relapsing armament of patient care. This primarily influenced the OS data. Another limitation has to do with the use of four-colored flow cytometry in this study and not the more accurate eight multi-colored flow cytometry [21]; also relevant is the fact that immunogold or mass spectrometry was limited to selected patients. However, the latter point may be advantageous for centers not utilizing next-generation flow.

Overall, however, we found CD24 to be the only antigen measured by MFC with prognostic significance for PFS and OS in patients newly diagnosed with MM and treated in the era of novel agents. Elevated CD24 expression correlated with better prognosis. This finding, combined with the known and validated prognostic factors, can provide a more extensive specific prognostic assessment profile of individual patients.

## Figures and Tables

**Figure 1 jcm-11-02913-f001:**
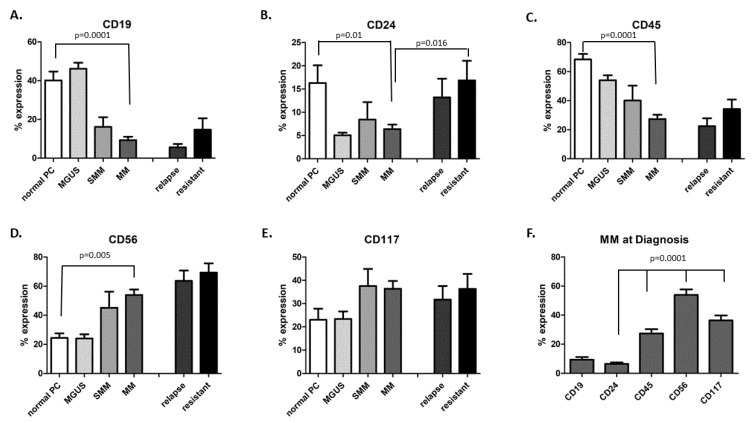
Flow cytometry level of CD expression on plasma cells at the different stages of MM development (**A**–**E**) and combined at diagnosis (**F**). Abbreviations—PC: plasma cells, MGUS: monoclonal gammopathy of undetermined significance, SMM: smoldering multiple myeloma, MM: multiple myeloma.

**Figure 2 jcm-11-02913-f002:**
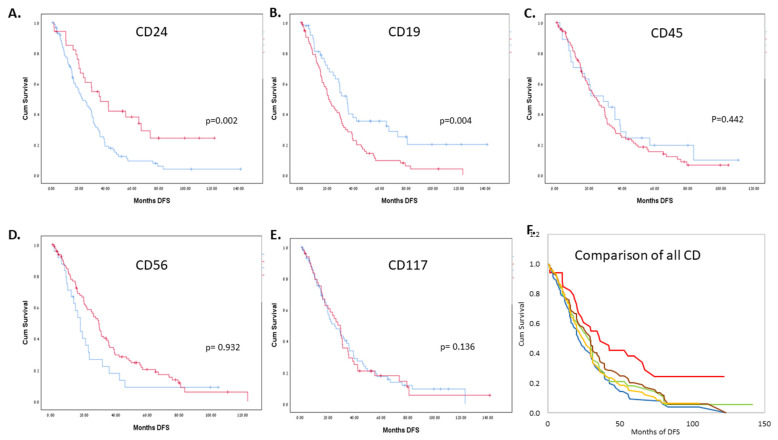
(**A**–**E**) Correlation of progression-free survival (PFS) to the CD levels measured by MFC at diagnosis, with a cutoff expression level of 5% on PCs (red above 5%, blue below 5%). (**F**) A visual comparison between the different CDs, pooled (**F**). All lines represent the expression of CD above 5%. CD24-red, CD117-brown, CD19-blue, CD45-yellow, CD56-green.

**Figure 3 jcm-11-02913-f003:**
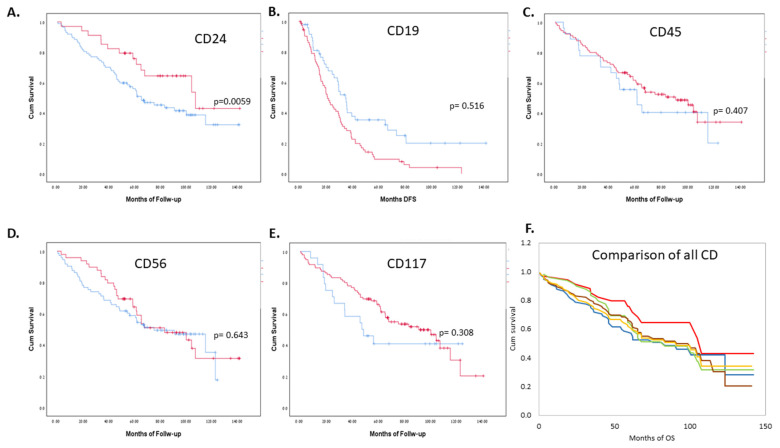
(**A**–**E**) Correlation of overall survival (OS) to the CD levels measured by MFC at diagnosis, with a cutoff expression level of 5% on PCs (red above 5%, blue below 5%). (**F**) A visual comparison of the different CDs, pooled (**F**). All lines represent the expression of CD above 5%. CD24-red, CD117-brown, CD19-blue, CD45-yellow, CD56-green.

**Table 1 jcm-11-02913-t001:** Patient baseline characteristics at diagnosis for the main cohort.

		AL	MM	CD24+ (above 5%)	CD24− (5% and below)	Total
Gender	Male	9 (47.4%)	66 (62.9%)	19 (55.9%)	54 (62.1%)	75 (60.5%)
	Female	10 (52.6%)	37.1 (37.5%)	15 (44.1%)	33 (37.9%)	49 (39.5%)
Median age at diagnosis (years, range)		68.5 (48–86)	62 (30–89)	64.5 (30–80)	63 (35–89)	62.5 (30–89)
M-protein (for MM patients)	IGG		54 (51.4%)	54 (43.5%)	18 (52.9%)	34 (39.1%)
	IGA		19 (18.1%)	19 (15.3%)	3 (8.8%)	16 (18.4%)
	LC only		30 (28.6%)	49 (39.5%)	13 (38.2%)	35 (40.2%)
	Non-secretory		2 (1.9%)	2 (1.6%)	0	2 (2.3%)
Light chain	Kappa	8 (42.1%)	64 (61.5%)	72 (58.0%)	14 (41.2%)	55 (63.2%)
	Lambda	11 (57.9%)	39 (37.5%)	50 (40.3%)	20 (58.8%)	30 (34.5%)
	Non-secretory	0 (0%)	2 (1.9%)	2 (1.6%)	0	2 (2.3%)
R-Mayo stage (for AL patients)	1	4 (21.0%)				
	2	1 (5.3%)				
	3	7 (36.8%)				
	4	7 (36.8%)				
R-ISS (for MM patients)	1		37 (35.2%)			
	2		36 (34.3%)			
	3		24 (22.9%)			
	Missing		8 (7.6%)			
Organ involvement (for AL patients)	Heart	14 (73.7%)				
	Kidney	13 (68.4%)				
	Soft tissue	11 (57.9%)				
	PNS	11 (57.9%)				
	Liver	3 (15.8%)				
	GI	1 (5.3%)				
Cytogenetics *	Standard risk		66 (62.9%)	26 (76.5%)	49 (56.3%)	78 (62.9%)
	High risk		24 (22.9%)	4 (11.8%)	22 (25.3%)	26 (21.0%)
	Missing		15 (14.3%)	4 (11.8%)	16 (18.4%)	20 (16.1%)
Calcium (above or below 11 mg/dL)	High	0 (0%)	7 (6.7%)	0	7 (8.0%)	7 (5.6%)
	Normal	17 (89.5%)	96 (91.4%)	31 (91.2%)	79 (90.8%)	113 (91.1%)
	Missing	2 (10.5%)	2 (1.9%)	3 (8.2%)	1 (1.1%)	4 (3.2%)
Creatinine (above or below 2 mg/dL)	Normal	16 (84.2%)	86 (81.9%)	27 (79.4%)	72 (82.8%)	102 (82.3%)
	High	2 (10.5%)	17 (16.2%)	5 (14.7%)	14 (16.1%)	19 (15.3%)
	Missing	1 (5.3%)	2 (1.9%)	2 (5.9%)	1 (1.1%)	3 (2.4%)
Lytic lesions (for MM patients) **	Yes		66 (62.9%)	14 (41.2%)	41 (47.1%)	57 (46%)
	No		39 (37.1%)	20 (58.8%)	46 (52.9%)	67 (54%)
Hemoglobin (above or below 10.5 g/dL)	Normal	14 (73.7%)	58 (55.2%)	25 (73.5%)	45 (51.7%)	72 (58.0%)
	Low	4 (21.0%)	45 (42.9%)	7 (20.6%)	41 (47.1%)	49 (39.6%)
	Missing	1 (5.3%)	2 (1.9%)	2 (5.9%)	1 (1.1%)	3 (2.4%)
Induction regime	VCD	19 (100%)	100 (95.2%)	31 (91.2%)	85 (97.7%)	119 (96.0%)
	VTD	0(0%)	5 (4.8%)	3 (8.8%)	2 (2.3%)	5 (4.0%)
ASCT	Yes	0 (0%)	57 (54.3%)	11 (32.4%)	44 (50.6%)	57 (46.0%)
	No	19 (100%)	48 (45.7%)	23 (67.6%)	43 (49.4%)	67 (54.0%)

Abbreviations—MM: multiple myeloma, AL: amyloidosis, IGG: immunoglobulin G, IGA: immunoglobulin A, LC: light chain, R-Mayo: revised Mayo stage, R-ISS: revised International Staging System, PNS: peripheral nervous system, GI: gastro-intestinal, VCD: bortezomib, cyclophosphamide, and dexamethasone, VTD: bortezomib, thalidomide, and dexamethasone, ASCT: Autologous stem cell transplantation.* High-risk cytogenetics: presence of del (17p) and/or translocation t (4;14) and/or translocation t (14;16). ** As identified in CT/PET-CT or MRI.

**Table 2 jcm-11-02913-t002:** Comparative prevalence of CDs in AL and MM (*p* value = NS for all variables).

	AL		MM	
	Median	Std. Deviation	Median	Std. Deviation
CD19 %	2.00	2.45	10.08	16.53
CD24 %	6.65	11.18	1.21	19.44
CD56 %	3.55	5.14	3.40	14.61
CD117 %	37.25	36.97	79.24	40.75
CD45 %	49.08	33.69	17.05	35.35
BM PC%	12.06	13.15	9.79	34.46

Abbreviations—AL: amyloidosis, MM: multiple myeloma, BM: bone marrow, PC: plasma cells. NS: non-significant.

**Table 3 jcm-11-02913-t003:** Comparative significance values of CDs correlated to MM and AL.

	MM			AL		
	Hazard Ratio	Confidence Interval	*p*-Value	Hazard Ratio	Confidence Interval	*p*-Value
CD 24%	0.980	0.959–1.002	0.075	0.900	0.799–1.013	0.082
CD 19%	1.009	0.993–1.025	0.287	1.199	0.959–1.498	0.110
CD 56%	0.998	0.980–1.016	0.821	0.911	0.786–1.055	0.212
CD 45%	0.995	0.987–1.003	0.224	0.995	0.977–1.012	0.552
CD 117%	0.999	0.992–1.006	0.795	0.994	0.978–1.010	0.442
PC%	0.984	0.973–0.995	0.003	0.983	0.924–1.047	0.600

**Table 4 jcm-11-02913-t004:** Correlations between the progression-free survival hazard ratio and CD expression levels as continuous variables.

CD	Hazard Ratio	Confidence Interval	*p*-Value
CD24	0.975	0.960–0.991	*p* = 0.002
CD19	1.025	1.014–1.037	*p* < 0.001
CD117	1.001	0.996–1.006	*p* = 0.770
CD56	1.003	0.990–1.016	*p* = 0.641
CD45	1.004	0.999–1.010	*p* = 0.120

**Table 5 jcm-11-02913-t005:** Correlation between overall survival hazard ratio and CD expression levels as continuous variables.

CD	Hazard Ratio	Confidence Interval	*p*-Value
CD24	0.979	0.958–0.999	*p* = 0.044
CD19	1.007	0.992–1.023	*p* = 0.362
CD117	0.998	0.992–1.005	*p* = 0.601
CD56	0.982	0.951–1.014	*p* = 0.255
CD45	0.995	0.988–1.003	*p* = 0.203

**Table 6 jcm-11-02913-t006:** Multivariate analysis showing the statistically significant factors correlating with poorer OS and CD24%.

	Hazard Ratio	Confidence Interval	*p*-Value
R-ISS	1.917	1.074–3.425	*p* = 0.028
High-risk cytogenetics *	3.452	1.719–6.9333	*p* < 0.001
High Β2M (mg/mL)	1.000	1.000–1.000	*p* = 0.044
Hypercalcemia **	3.642	1.637–8.105	*p* = 0.002
Older age	1.047	1.014–1.081	*p* = 0.005
CD24%	1.011	0.986–1.037	*p* = 0.383

Abbreviations—OS: overall survival, R-ISS: revised International Staging System, Β2M: beta 2 microglubolin. * High-risk cytogenetics: presence of del (17p) and/or translocation t (4;14) and/or translocation t (14;16). ** Hypercalcemia: above 11 mg/dL.

## Data Availability

Data supporting the results can be obtained from the corresponding author.

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
