# Peer review of "CD24 Is a Prognostic Marker for Multiple Myeloma Progression and Survival"

_jcm, 2022, doi:10.3390/jcm11102913_

Round 1

Reviewer 1 Report

The study demonstrate the expession of CD24 correlated with the PFS in plasma cell disorder both MM and AL amyloidosis. The author divided patients characteristic and analyze the data separately of both cohorts which have different clinical course and prognosis.

I have a suggestion on sentence 281 - 284
-Furthermore, the tests to calculate the R-ISS (i.e. FISH) are expensive and not universally available [37]. Other known prognostic factors were not included in the R-ISS algorithm [38,39]. In this respect, MFC CD24 remains an important additional modality to assess prognosis in the newly diagnosed MM patient.-

I add some points about FCM. The use of CD 24 could be able to be a prognositc factor. However, the FCM is not a madatory test for diagnosis MM. Moreover, CD24 is not in the standard use of FCM in plasma cell disorder.

Reviewer 2 Report

Even-Zohar provide a well-written publication about CD24 as prognostic factor MM. The quality of presentation is high, both introduction and discussion are sound. There are a few questions regarding data analysis which I would like to be adressed:

Please include the time between diagnosis and sampling in table 1.

Please include the exact induction regimen (VCD? VRD?) in table 1.

Why was 1q21 not considered as high-risk cytogenetic marker?

Autologous transplantation prolongs PFS in NDMM. Please show Kaplan-Meier curves according to CD24 expression for the transplant and non-transplant group separately (as supplement).

Table 6 (multivariate analysis) should include CD24 as marker in order to show wether CD24 is an independent prognostic marker. It could well be that the prognostic impact is due to higher prevalence of standard-risk cytogenetics (76.5% vs. 56.4%) in CD24+ patients. If CD24 does not significantly correlate in multivariate analysis, please comment on that in the discussion

Please include 2-3 sentences on wether CD24 could be utilized as therapeutic target in MM.

Author Response

This manuscript is a resubmission of an earlier submission. The following is a list of the peer review reports and author responses from that submission.

Round 1

Reviewer 1 Report

The major concerns are:

1) no confirmation of clonality is reported in the flow panels (no light chains) so it is not known if data are referred to monoclonal or polyclonal plasma cells

2) MM and amyloidosis patients are considered together but the outcome is different

3) ASCT and age should be included in the evaluation of the multivariate model

4) table 1 should be divided in CD24 positive or negative patients and authors should make a choice between cosidering CD24 a continuous or categorical variable with a cutoff justified by data.

Reviewer 2 Report

This study demonstrated important data of CD24 expression on plasma cell disorder which might be beneficial for detemining prognosis. I have some points to suggest.

1. In the method session
1.) Please indicated the data from MGUS, normal, and at relapse, refractory stages. Is this a sequencial sample collection or only 124 patients. How you collect the sample from normal and MGUS. Is there the inform consent from these pateints?

2.) How you confirm the diagnosis of AL amyloidosis?

3.) Please give more information on the imaging method that using in identification bone lesions.

4.) Please define the high-risk cytogenetics.

2. In this cohort included patients both MM and AL amyloidosis
1.) AL amyloidosis have differnt prognosis factors when compare with MM. Is there any information of CD24 expression and other CD that you included in this study that predict the prognosis of the disease. Could you plesase explain more information in the introduction part?

3. Result
1.) Please demonstrate the patients characteristic before the expression of each CD (line 114 - 122). 

2.) Could you demonstrate the patients characteristic of AL amyloidosis such as organ involvement (cardiac, renal), staging of amloidosis which could better help the reader intepret the survival outcome. Although, they have similar CD expression, the R-ISS staging did not predict survival outcomes in amyloidosis.

3.) Although, amyloidosis cohort express similar CD to MM, the response to the treatment and survival outcome are different. I suggest to separate the result or include only MM patients.

4. Discussion: 
1.) mostly explain the important of CD24 in MM. There was no information in AL amyloidosis. Could you explain more correlation between CD24 and prognosis. 

2.) Please add the limitation of the study.

Round 2

Reviewer 1 Report

1) No confirmation of clonality is reported in the flow panels (no light chains) so it is not known if data are referred to monoclonal or polyclonal plasma cells

We thank the reviewer for bringing this point. It is not common practice to determine clonality of PCs by flow at diagnosis unless the immunophenotype using the 8 antibody panel does not clarify the aberrant versus the normal PCs. In most cases expression of CD45, CD19, CD56 and CD117 together with CD27 and CD81 is enough for determining the aberrant PC percent in 99% of cases. If the sample monoclonality was not clear Kappa/Lambda intracellular staining was performed. We have added this to the methods section, line 104-105. Furthermore, the expression of CD24 for these patients was not correlated to the amount of aberrant PCs, but to total PCs present. As these were newly diagnosed active MM patients, in all cases more than 95% of PCs were aberrant in the patients in this study.

In how many patients clonality was performed? Moreover you state that the expression of CD24 for those patients was not correlated to the amount of aberrant PCs, but to total PCs however more than 95% of PCs were aberrant. Where do you think the role of CD24 on prognosis resides ? On the monoclonal or polyclonal PCs? It is not clear to me.

2) MM and amyloidosis patients are considered together but the outcome is different

We agree with this statement. MM and amyloidosis are different diseases with different outcomes. In this specific cohort in whom all patients were treated with the same protocol, the OS and PFS were found to be similar.  The expression levels of the different CDs were similar as well. The fact that we combined those two different groups allowed us to enlarge the statistical power of the group and to better understand the effect of CD24. In order to clarify this important point we attached the relevant figure with the OS of MM and AL and a table of the expression of the different CDs in AL and MM, to the supplementary material showing this (figure 1 and table 1). We have made further clarification in the results section- lines 158-163.

Were the patients with AL amyloidosis also diagnosed with active Myeloma? The 2 conditions may coexist

4) Table 1 should be divided in CD24 positive or negative patients and authors should make a choice between considering CD24 a continuous or categorical variable with a cutoff justified by data.

We changed table 1 according to this recommendation. We considered CD24 as a categorical variable with a 5% cutoff as explained in the materials and methods section.

In the manuscript table 2 is still referring to CD24 as a continuous variable as the title state. The same in line 179-182. 

Moreover, In table 2 of the supplementary hypercalcemia have a big impact (usually  while LDH is not analysed (that is part of the R-ISS). Why? Moreover high B2M should be evaluated with albumin in the context of ISS.

Reviewer 2 Report

This study contained 2 cohorts, first cohort represent the CD expression which included normal subjects. The second cohort included newly diagnosed MM and AL amyloidosis for CD expression and survival. It is correct that the cohort for the purpose of CD24 expression in different disease states are informed consent.

In the edited version could improve in some points. But it still have major pitfalls. Such as the cut-off for the CD expression at 5% for CD expression, the inlucing of AL amyloidosis in the PFS analysis for larger cohort which could lead to the misinterpretation the results. 

The patients' characteristics of bone lesions that included all 124 patients. However, the bone lesion should indicate only MM patitents becuase AL amyloidosis should not have lytic lesions. Otherwise these patients were diagnosed as MM with amyloidosis which would have diffent prognosis.

The statement: "No data regarding CD24 in AL exists to our knowledge. The fact that AL, like MM, is a plasma cell disorder treated by similar protocols may be the ground to better understanding the common mechanisms in which CD24 effect the prognosis in AL too". Currently, we have the updated data that orgination of AL amyloidosis is different from MM. Also, although, these 2 diseases use almost similar regimen, the treatment consideration and dose adjustment are totally different. Moreover, the response to the treatment are also different.

The limitations that added in the discussion is good. Moreover, the immunohistochemistry typing is not s gold standard pf doagnosis. This should be added in the limitation.